

# Analysis of Snow Cover Changes Using MODIS Snow Products and Meteorological Data in the Hunza Region, Karakoram, Pakistan

Muqeet Ahmad [1,2], Shehzad Ali [3], Garee Khan [2,] Parisa Karim [1,2], Muzammil Hassan [1,2] Kelden Jurmey [1], Lhachi Dema [1], Shrija Gurung [1]

[1] Department of Environmental Science and Engineering, Kathmandu University, Dhulikhel 45200, Nepal
[2] Department of Environmental Science, Karakoram International University, Gilgit-Baltistan, Pakistan
[3] Department of Geography, Government College University, Lahore, Pakistan
Corresponding to: muqet100@gmail.com

**Abstract**: The Upper Indus Basin (UIB) is characterized by contrasting meteorological behaviors; therefore, it has
become pertinent to understand the meteorological trends at the sub-basin level. Many studies have investigated the
snow-covered area along with meteorological trends at the basin level. Still, none have reported the spatial
variability of trends and their magnitude at a sub-basin level. This study is conducted to monitor the seasonal trends
in the snow-covered area and climatic factors (temperature and precipitation) in the Hunza region of the Upper Indus
Basin (UIB). Summer and winter seasons were selected because temperature and precipitation during these two
seasons are the key factors for snow cover variation in the region. Mann-Kendall and Spearman methods were used
to study the seasonal trends and their magnitude using MODIS snow cover information (2000–2020) and
meteorological data. The results showed that during the summer season, SCA and precipitation showed a non-
significant ($p=0.05$) decreasing trend with a value of -0.0095 km²/month and -0.191 mm/month, respectively, while
the temperature showed a non-significant increasing trend with a value of 0.315°C/month. While, during the winter
season, SCA and temperature showed a non-significant increasing trend with a value of 0.114 km²/month and
0.176°C/month, respectively, and precipitation showed a significant increasing trend with a value of 0.436
mm/month. In general, the snow-covered areas of the Hunza region have an increasing trend during the winter
season, while the summer season has a decreasing trend of snow-covered areas. Based on the results of this study it
can be concluded that since the Hunza sub-basin of the UIB is influenced by a different climatological system
(westerly system) as compared to other sub-basins of the UIB (monsoon systems), the results of those studies that
treat the UIB as one unit in meteorological modeling should be used with caution. Furthermore, it is suggested that
similar studies at the sub-basin level of the UIB will help in a better understanding of the Karakoram anomaly.
**Keywords:** Upper Indus Basin (UIB); Snow-Covered Areas (SCA); Hindu-Kush Himalayas (HKH), MODIS;
Mann-Kendall trend analysis.



## 1.    Introduction

Snow Cover is one of the most important land surface characteristics in terms of the global energy budget and the
water cycle (Ahmad et al., 2018). The world population is heavily reliant on melt-water resources from glaciers and
snow (Lutz et al., 2014). According to Bormann et al. (2018), 17% of the world's population relies on seasonal
glacier melt and snowmelt water. According to (Ghulam Rasul, 2011), temperatures are projected to increase
globally by 0.6°C between 2001 and 2010. However, the actual temperature rise was 0.93°C, and the increase was
even greater in northern Pakistan, where the temperature increased by 1.3°C. As a result, glaciers in the HKH region
have retreated and snow has melted. The recent climate warming puts these natural resources at risk; according to
the Intergovernmental Panel on Climate Change (IPCC) report titled "State of the Global Climate 2020 WMO-No.
1264", between 1880-2020 where the average land and ocean temperature has increased globally by 1.2°C.
The Hindu-Kush Himalayas (HKH) region, also known as the "Water Towers of Asia", provides natural freshwater
reserves in the form of glaciers and snow, which ensure water distribution in the surrounding mountainous areas and
the adjacent plains. The hydrology of the HKH region is primarily driven by glaciers and snowmelt, which sustains
rivers and streams in the area (Immerzeel et al., 2010). Furthermore, the pace of glacier retreat in the Himalayan
region has increased dramatically in recent decades, which is considered to be a result of global warming
(Dyurgerov and Meier, 2005; Ageta et al., 2000). The majority of the glaciers and snow cover in the HKH region is
found in Pakistan's Indus Basin catchment, with a total glacier area of 21,192.67 km² (Bajracharya and Shrestha,
2011). Pakistan's Indus irrigation network is one of the largest irrigation systems that are vital to Pakistan's economy
(Tahir et al., 2015). The agricultural sector is the backbone of Pakistan's economy, which is heavily dependent on
water flow in the Indus Irrigation System (IIS) which gets 80% of its inflow from the perennial snow-covered areas
of HKH region. The major Tarbela reservoir of the IIS, receiving its primary input from glaciers and snowmelt from
the Upper Indus Basin (UIB), provides water to Punjab and Sindh's agricultural areas (Archer et al., 2010). The UIB
covers about 200,000 km² area (Atif et al., 2018; Hewitt, 2005) and is split into five sub-basins: Gilgit, Astore,
Shigar, Shyok, and Hunza, which acquire around 60% of the meltwater from snow-covered areas (SCA) and glaciers
(Tahir et al., 2015). Because the people living downstream rely heavily on these water resources and it's critical to
understand how these basins water resources behave.
The volume, distribution, and extent of snow are all affected by rising temperatures, making mountain hydrology
more sensitive to large-scale changes (Laghari et al., 2012). According to recent studies, snow cover has declined
during the previous four decades, particularly in the northern hemisphere (Brown and Robinson, 2011; Choi et al.,
2010; Kunkel et al., 2016). In the western United States, a declining trend of snow-covered areas (SCA) was
detected based on ground measurements (Pederson et al., 2010; Harpold et al., 2012). Additionally, due to rising
temperatures, snowfall across the Himalayan region decreased by approximately 16% from 1990 to 2001 (Menon et
al., 2010). Immerzeel et al. (2009) the same pattern was observed between 2000 and 2008. Many scientists say that
the snow cover area (SCA) in the HKH region is decreasing (Immerzeel et al., 2009; Gurung et al., 2011b; Shrestha
and Joshi, 2011; Maskey et al., 2011; Gurung et al., 2011a), On the other hand, increased SCA has been reported in
the western Himalayas and Karakoram (Tahir et al., 2015). Snow cover has also dropped across the Arctic,
according to satellite data (Shi et al., 2013). Under global warming, the global snow cover is one of the rapidly



changing hydrologic systems, with wide implications for ecological function (Allan and Soden, 2007). Because of
the predicted rapid snow cover retreat hindered by the shift of snowmelt runoff from the summer to the winter
season, there is a greater risk of winter floods and summer droughts (Ahmad et al., 2018).
The Snow-Covered Areas (SCA) of the Hunza region of the UIB have recently received a lot of attention. Many
researchers (Bilal et al., 2019; Tahir et al., 2011) utilized meteorological variables along with the satellite data of
different spatial and temporal resolutions to carry out the trend analysis of SCA variability of the region. These
studies, however, did not take into account the variability of meteorological factors at the Sub-Basin level of the
region. Moreover, because it is impossible to establish an observatory in a whole area, meteorological stations offer
ground data with a restricted coverage area, therefore, using Remote-Sensing Data is the sole option to evaluate the
changes in snow cover in remote locations like the Upper Indus Basin (UIB) (Tahir et al., 2015). However, the
investigation was hampered by a lack of data at the high altitude and the availability of data at low altitude in the
Hunza region. Many studies have raised this concern (Atif et al., 2018; Bilal et al., 2019; Tahir et al., 2011). The
MODIS/Terra Snow Cover 8 Day L3 Global 500 m Grid (MOD10A2) V006 was used to monitor snow cover for 20
years from 2000 to 2020. The nonparametric Mann-Kendall and Spearman statistical techniques were used to
determine the trend and magnitude of the meteorological variations (temperature & precipitation) in the Hunza
region and its correlation with SCA.

## 2. Materials and Methods

### 2.1. Study Area.

The Hunza region of the UIB (8,732.4 km²) is situated in the Gilgit-Baltistan region of Northern Pakistan (Jamill et
al., 2022) within a high mountainous area between 74°02′ and 76°00′ East and 35°54′ and 37°10′ North. Hunza is
bordered to the Northeast by China's Xinjiang region, to the Northwest by Afghanistan, and to the South by the
Nagar district (Nagar 1 in the Southeast and Nagar II in the Southwest). The Water and Power Development
Authority (WAPDA), under the Pakistan Snow & Ice Hydrology Project (PSIHP), has deployed two climate stations
with precipitation gauges at different elevations: Khunjerab at 4,730 m asl and Ziarat at 3,669 m asl. The climate in
Hunza is dry to semi-arid, with two main seasons, Summer from April to August and Winter from September to
March. According to 20 years of records (2000–2020) from these climate stations, the annual mean precipitation is
180 mm at Khunjerab (4,730 m) and 225 mm at Ziarat (3,669 m) (Atif et al., 2018). Nearly half of the basin area lies
above 4,500 m in elevation. Figure 1 shows the Hunza area and drainage basin along with the meteorological
Stations, while Table 1 details the basin's key physical characteristics.



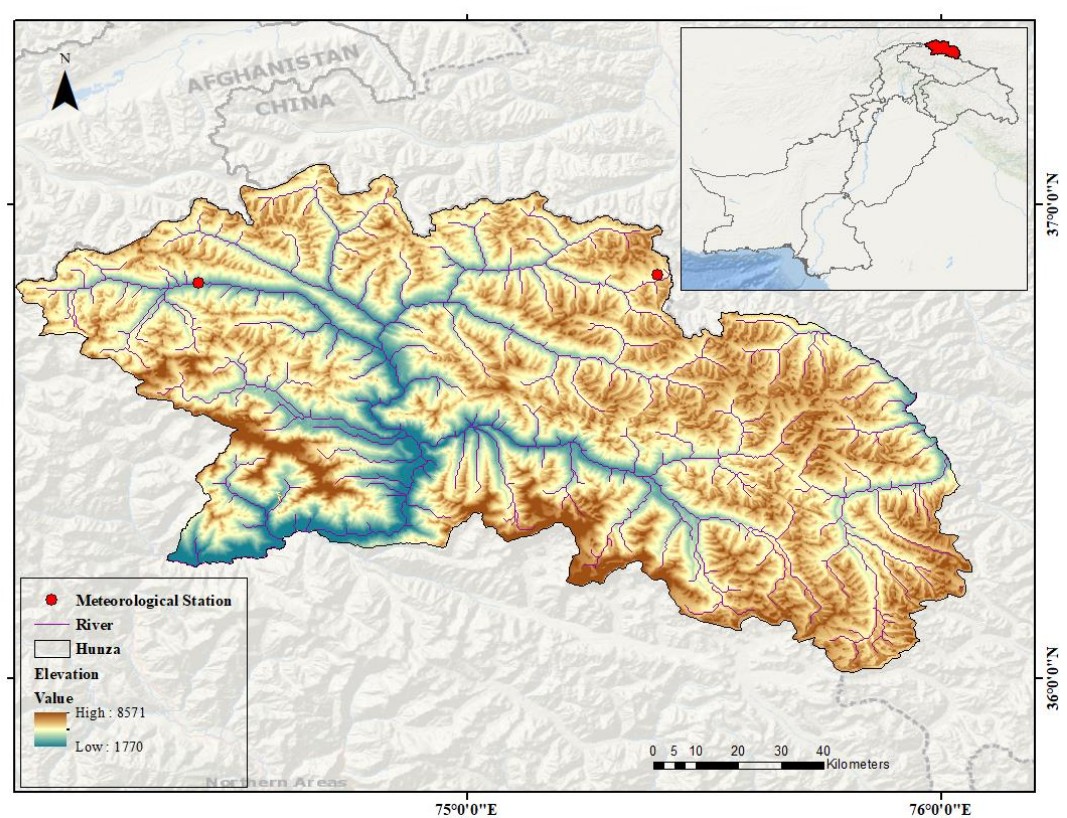

Figure 1 Map of the study area along with the installed meteorological stations.

| Physical Characteristics | Description |
|---|---|
| Catchment area | 8.732 km² |
| Elevation range | 1,774-7,774 m |
| Mean elevation | 4,774 m |
| Latitude | 35°54' to 37°10' |
| Longitude | 74°02' to 76°00' |
| Catchment area above 4500m | 62.40% |
| Dominant land use type | Bare soil with shrubs |
| **Mean Precipitation (2000-2020)** | |
| Khunjerab (4730 m) | 163 mm |
| Ziarat (3669 m) | 292 mm |





86         Table 1 Major physical characteristics of the Hunza region.

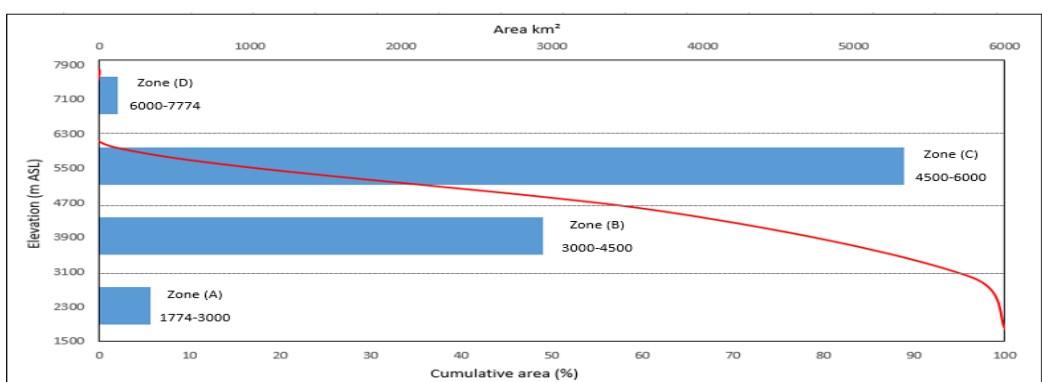

87     Figure 2 Hypsometric curve of Hunza region estimated through digital elevation model (DEM).

| Name | Elevation range (m asl) | Mean elevation (m asl) | Land area ( km$^2$) | Land area % |
|---|---|---|---|---|
| **Zone (A)** | 1774-3000 | 2387 | 342.24 | 3.92 |
| **Zone (B)** | 3000-4500 | 3750 | 2937.87 | 33.64 |
| **Zone (C)** | 4500-6000 | 5250 | 5328.04 | 61.01 |
| **Zone (D)** | 6000-7774 | 6887 | 124.28 | 1.42 |

88      Table 2 Summary of the hypsometric zones along with the land area percentage.

### 2.2 Methodology

89 This study has faced challenges due to the lack of data at high altitudes and the availability of data at low altitudes in

90 the Hunza region. Many studies have raised this concern (Atif et al., 2018; Bilal et al., 2019; Tahir et al., 2011). High

91 winds have been observed to cause the underestimation of solid precipitation at high altitudes (Dahri et al., 2016;

92 Ménégoz et al., 2013). However, the MODIS 8-day snow product was used to examine snow cover during 20 years

93 from 2000 to 2020. To identify Snow-Covered Areas across different zones, a study was conducted from 2000 to

94 2020 using supervised classification of Landsat images with a 30m spatial resolution, classifying them into five

95 categories: snow, ice, vegetation, rock, and soil. This was followed by manual digitization to refine snow boundaries

96 and remove artifacts. The nonparametric Mann-Kendall and Spearman statistical techniques were used to determine

97 the trend and magnitude of the meteorological variations in the Hunza region and its association with the SCA. The

98 arithmetic mean of the daily maximum and minimum temperatures is used to get the daily average temperature.

99 Unrecorded data was replaced by interpolating data from the previous and subsequent years. Figure 3 illustrates the

100 Methodological framework of the study.



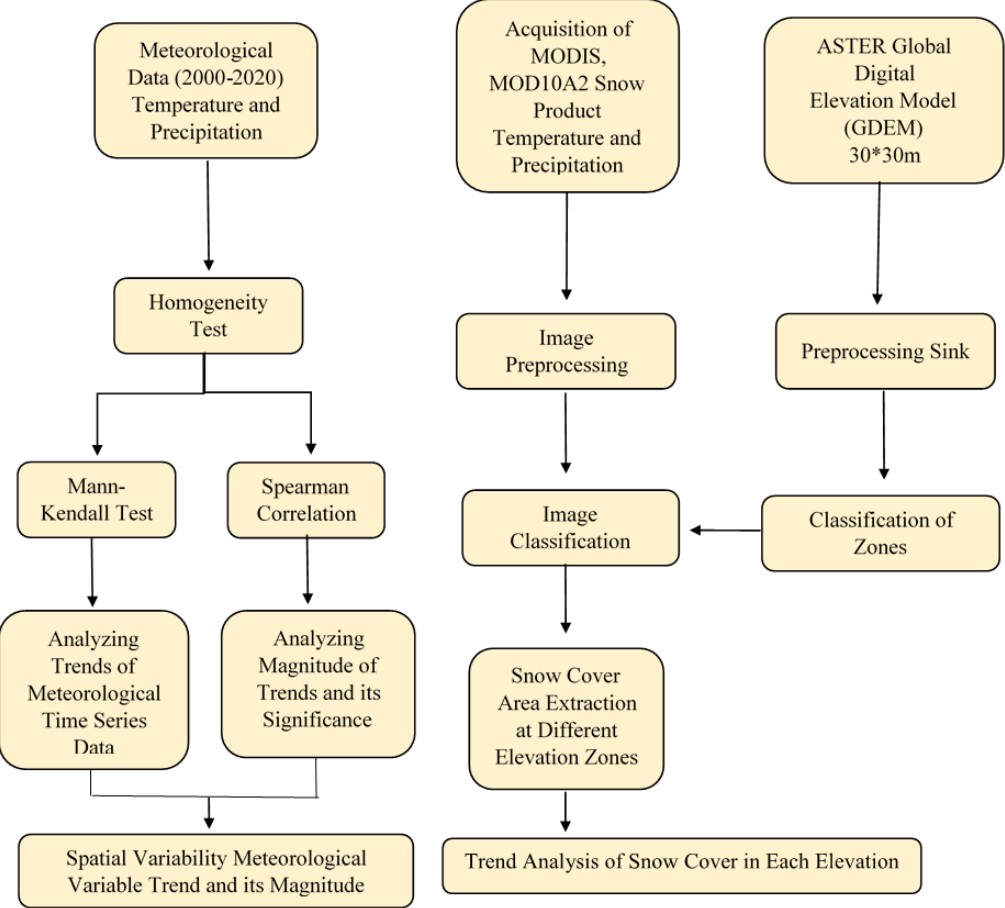

Figure 3 Methodological framework of the study.

**Homogeneity Test**

The Standard Normal Homogeneity Test (SNHT) was used to evaluate the homogeneity of hydrometeorological and
snow cover data at a 5% significance level for each station (Kang and Yusof, 2012). Data is considered to be
homogeneous when the computed $p$-value is less than 0.05 (Atif et al., 2018). All the meteorological variables were
found to be homogenous.

**Trend Analysis**

For Seasonal Trend Analysis, climatic variables from both these stations were divided into two seasons: Winter
(March) and Summer (August), because the trend in temperature and precipitation during these two seasons are the
key factors for Snow Cover Variation in the Hunza region (Dimri et al., 2013; Tahir et al., 2015). To find patterns in



time series data of the meteorological variables, a non-parametric Mann- Kendall trend test for the seasonal data was
used. The MK statistic reveals the nature of a trend, whether it is positive or negative (Mann, 1945; Kendall, 1975).
It does not, however, give information on the Slope's Magnitude or the trend line. The size of the trend was
estimated using the Spearman correlation method at a 5% significance level ($\rho=0.05$).
The mathematical equations for calculating Mann-Kendall Statistics (S) are as follows:

$$S = \sum_{i=1}^{n-1} \sum_{j=i+1}^{n} sign(x_j - x_i) \tag{1}$$


Note that if $S > 0$ then later observations in the time series tend to be larger than those that appear earlier in the time
series, while the reverse is true if $S < 0$.
The variance of $S$ is given by

$$var = \frac{1}{18}\left[ n(n-1)(2n+5) - \sum_{t} f_t(f_t - 1)(2f_t + 5) \right] \tag{2}$$


Where $t$ varies over the set of tied ranks and $f_t$ is the number of times (i.e. frequency) that the rank $t$ appears.
The MK Test uses the following test statistic:

$$z = \begin{cases} (S-1)/se, & S > 0 \\ 0, & S = 0 \\ (S+1)/se, & S < 0 \end{cases}$$


**2.3 Data Sets**

**Topographical Data**.

The Hunza region is defined using the using the Shuttle Radar Topography Mission (SRTM) Digital Elevation
Model (DEM) at a 30-meter resolution (1 arc-second). The STRM DEM was obtained from Earth Explorer and
provides detailed elevation data for the Hunza region, though some voids-filled higher elevation data at a resolution
of 30-meter resolutions. To address these voids in the STRM DEM were filled using the fill function in ArcMap 10.5.
The data was then divided into four distinct elevation zones, each with a mean elevation difference of about 1500
meters. The mean elevation of the Hunza basin, determined using an area-elevation hypsometric curve, was
calculated to be about 4,774 meters above sea level. This zoning helps in analyzing how the region's topography



affects snow cover variations. Figure 2.2 depicts the elevation range and proportionate area (percentage) of each
separate zone (A, B, C, and, D).

**Snow Cover Data.**

The MODIS/Terra Snow Cover 8 Day L3 Global 500 m Grid (MOD10A2) V006 was chosen to assess the spatial
variability of snow. This dataset was chosen for the study above other data sets because of its appropriate spatial and
temporal resolution, as evidenced by prior investigations (Ahmad et al., 2018; Tahir et al., 2011). These images
cover a land area of 1200 km by 1200 km and are projected using a sinusoidal map projection (Riggs et al., 2006).
The images were projected to the WGS 1984 UTM Zone 43N system in ArcGIS 10.5. To create a monthly
composite, the 8-day interval images were combined using ArcGIS's Composite tool. Image quality was further
enhanced using interpolation with the Inverse Distance Weighting (IDW) method in ArcGIS. After that, the "extract
by mask" tool of ArcGIS was used to demarcate the boundary of the study area and retrieve the area of interest
(AOI).

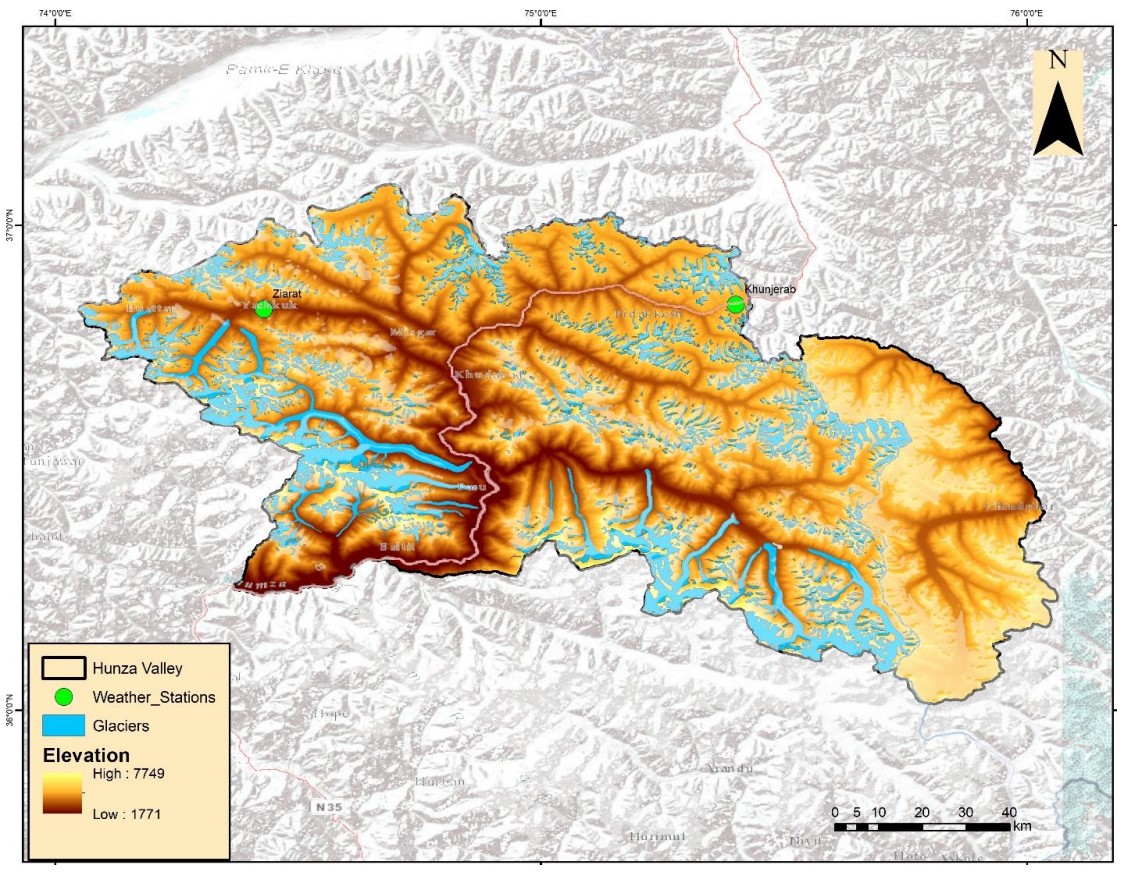



Figure 4 shows the altitudinal zones of the Hunza extracted from the Global Digital Elevation Model (GDEM) along
with glacier coverage and gauging stations.

**Meteorological Data**

Under the Pakistan Snow & Ice Hydrology Project (PSIHP), the Water & Power Development Authority (WAPDA)
made daily temperature, precipitation, relative humidity, and solar radiation data accessible for the two
meteorological Stations (Khunjerab and Ziarat) from 2000 to 2020. The Khunjerab station is located at an elevation
of 4730 m while the Ziarat station is at an elevation of 3669 m.

| Data | Spatial Resolution | Temporal Resolution | Source |
|---|---|---|---|
| **DEM** | Grid (30 m) | Fixed | SRTM |
| **Meteorological data** | Point | Daily | Water and Power Development Authority (WAPDA) |
| **Snow cover** | Grid (500 m) | 8-day maximum snow extent | MODIS Terra (MOD10A2) |

Table 3 represents the summary of the datasets utilized in the research.

### 3. Results and Discussion

#### 3.1. Climatic Trend Analysis

**Temperature Trend Analysis**

The months of April-August and September-March were assessed for temperature analysis, since both are classified
as two seasons (summer and winter, respectively) in the Hunza region, from 2000 to 2020. During the summer
season, the mean monthly temperature at the Khunjerab station is recorded at 10.22 °C, while the temperature
recorded at the Ziarat station is 24.09 °C. While during the winter season, the mean monthly recorded temperature at
Khunjerab and Ziarat stations is -4.39°C and 2.69 °C, respectively. Figure 5 represents the trend of mean
temperature for both summer and winter seasons from 2000 to 2020. From the figure, it could be seen that the
temperature for both the summer and winter has an increasing trend.



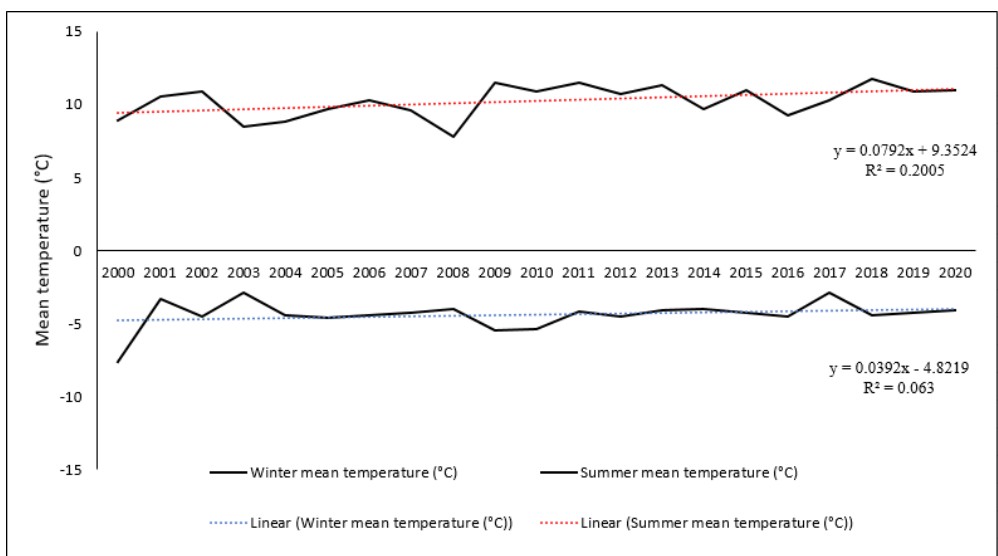

Figure 5 Temperature trend in the Hunza region during both summer and winter seasons from 2000 to 2020.

The Mann-Kendall (MK) test shows a minute, non-significant increasing trend in winter average temperature, with a
rate of 0.176 °C/month, while, on the other hand, the average summer temperature also shows a non-significant but
more notable trend as compared to winter temperature, with a rate of 0.315 °C (Table 4). From this result, it could be
concluded that, at a seasonal scale, the summer season shows the highest rate of temperature change followed by the
winter season. Atif et al. (2018) observed an overall increasing trend in average temperature at the hunza catchment.

|        | Trend      | p-value | Slope °C/month |
|--------|------------|---------|----------------|
| **Winter** | Increasing | 0.287   | 0.176          |
| **Summer** | Increasing | 0.052   | 0.315          |

Table 4 Summary of temperature trend analysis during both winter and summer sessions.

**Precipitation Trend Analysis**

During the summer season, the monthly precipitation at Khunjerab station was observed at 29.03 mm, while the
Ziarat station recorded monthly precipitation at 24.04 mm. On the other hand, during the winter season, monthly
precipitation recorded at Khunjerab station was 15.03 mm and 20.49 mm at Ziarat station, respectively. Figure 6
represents the general trend of monthly precipitation for both summer and winter seasons from 2000 to 2020. The
trend analysis shows that the summer precipitation is gradually diminishing concerning time while the winter
precipitation has an increasing trend from the period 2000 to 2020. According to (Hasson et al., 2014), the mean





annual precipitation at Khunjerab and Naltar stations within the Karakoram ranges from 200 to 700 mm,
respectively.

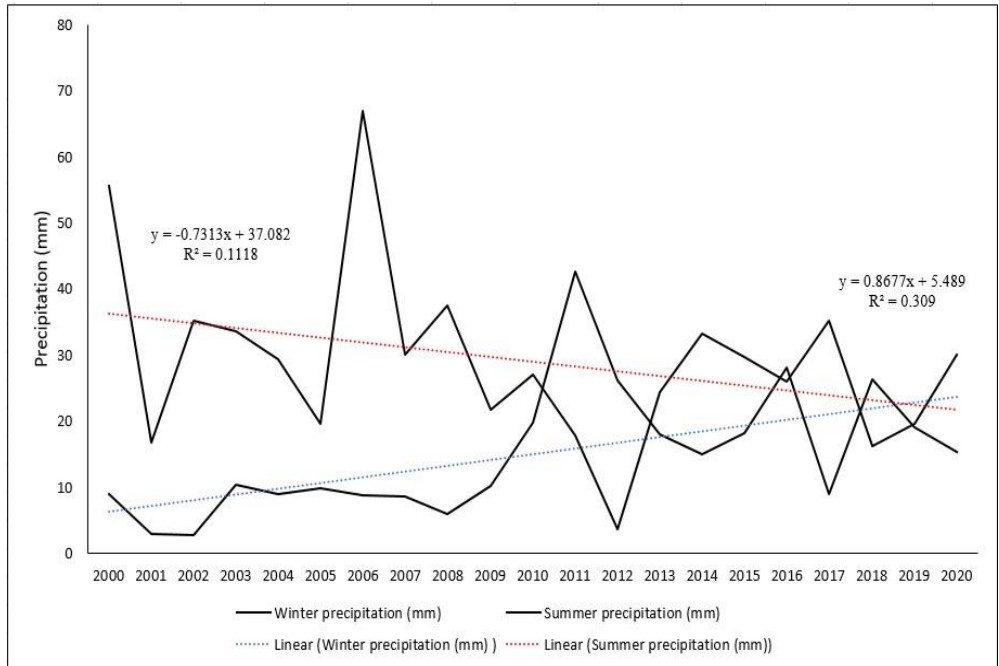

Figure 6 Precipitation trend in the Hunza region during both summer and winter seasons from 2000 to 2020.
The MK (Mann-Kendall) analysis of precipitation over the Hunza region reveals a significant increasing trend in
winter precipitation with a rate of 4.36 mm/month, and decreasing summer precipitation with a rate of -1.91
mm/month during the study period (2000-2020) (Table 5). Bilal et al. (2019) and Atif et al. (2018) Seasonal SCA in
the UIB is highly dependent on winter precipitation; consequently, an Increasing trend is observed in the
precipitation data. According to Tahir et al. (2011), a similar trend of increasing snow cover has been seen in the
Hunza basin due to increased winter precipitation. From 1961 to 2000, the Karakoram Mountains are reported to
have experienced a decrease in summer temperatures and a significant increase in precipitation (Fowler and Archer,
2006).

|  | Trend | p-value | Slope (mm/month) |
|---|---|---|---|
| **Winter** | Increasing | 0.006 | 0.436 |
| **Summer** | Decreasing | 0.238 | -0.191 |

Table 5 Summary of precipitation trend analysis during both winter and summer seasons.

### 3.2. Snow Cover Analysis of the Hunza Region





**Snow Cover Dynamics during Summer Season**

Analysis for the summer season shows that the average summer SCA (maximum extent) from 2000 to 2020 was
5630.83 km², accounting for 64.4% of the entire study region. During the summer season, the years 2014, 2004, and
2009 showed relatively more snow cover, with 6446.59 km², 6255.18 km², and 6138.27 km² in an area on average,
respectively of which SCA in 2014 showed the maximum snow cover, with 815.76 km² more area than the overall
average (2000-2020). Figure 7 represents the spatial variability of SCA during the summer season from 2000 to
2020.



Figure 7 Spatial variability of snow cover during the summer season from 2000 to 2020.

The trend analysis of SCA during the summer season is shown in Figure 8 SCA analysis during the summer season
represents a non-significant declining trend for the period of 2000 to 2020. Tahir et al. (2011) the Snow cover area is
at a minimum 30–40 % in the summer (July to September). This declining trend in SCA is also observed in the
previous studies conducted for the Hunza region (Bilal et al., 2019).

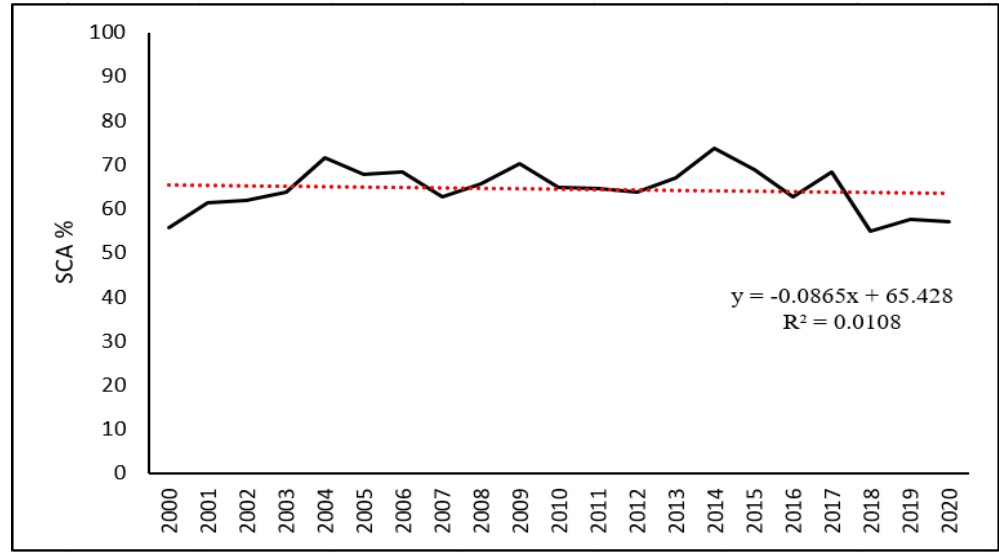

Figure 8 Trend analysis of snow-covered area during the summer season (2000-2020).

**Snow Cover Dynamics during Winter Season**

Analysis for the winter season shows that the average winter SCA (maximum extent) from 2000 to 2020 was
8126.61 km², accounting for 93% of the entire study region. During the winter season the years 2009, 2005, and
2018 showed relatively more snow cover, with 8419.89 km², 8338.28 km², and 8355.22 km² in an area on average,
respectively of which SCA in 2009 showed the maximum snow cover, with 293.28 km² more area than the overall
average (2000-2020). Figure 9 represents the spatial variability of SCA during the winter season in the Hunza region
from 2000 to 2020.





Figure 9 Spatial variability of snow cover during the winter season from 2000 to 2020.

The trend analysis of SCA during the winter season is shown in Figure 10 SCA analysis during the winter season represents a minute, non-significant increasing trend for the period of 2000 to 2020.



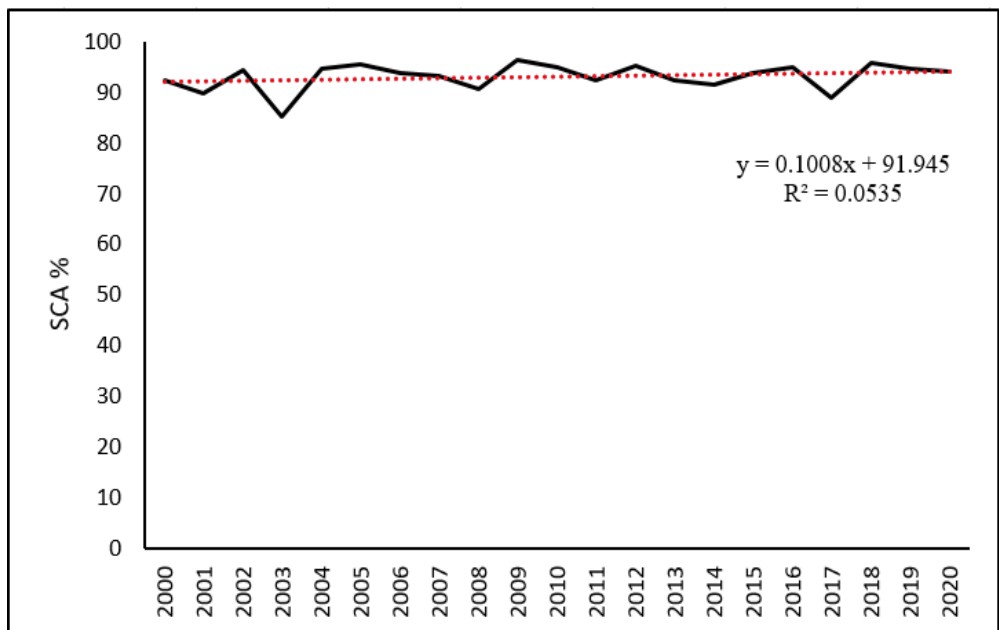

Figure 10 Trend analysis of snow-covered area during the winter season (2000-2020).
The Mann Kendall (MK) test shows a statistically significant increasing trend in winter SCA, with a rate of 0.114
km²/month. About 98% of the area of the Hunza basin lies above an elevation of 2000 m and about 32.5% of the
Hunza basin area is located above 5000 m where the temperature remains well below freezing throughout the year,
and which receives comparatively more snow due to the westerly system at this altitude. This could be the reason for
the slight increase in snow cover and the advancement of some glaciers in the Hunza basin (Atif et al., 2018).
According to Tahir et al. (2011), a similar trend of increasing snow cover has been seen in the Hunza basin due to
increased winter precipitation. 32.5% of the Hunza Basin region is located above 5,000 meters above sea level, and
the seasonal variation in snow cover ranges from 38 to 43%. At this altitude, temperatures remain below freezing
throughout the year and precipitation tends to increase. This may be the reason for the increase in snow cover in the
Hunza basin and the advance of some glaciers. (Hewitt, 2005; Archer and Fowler, 2004) also concluded that the
increase in snow cover and the advancement of glaciers is due to the increase in winter precipitation due to the
influence of westerly winds. While, on the other hand, a decreasing trend in summer SCA is observed at a rate of -
0.0095 km²/month (Table 6). A positive SCA trend and a decreasing temperature trend during the winter season and
a negative SCA trend and an increasing temperature trend during the summer season have also been observed in
previous studies (Bilal et al., 2019; Tahir et al., 2011).

|  | Trend | p-value | Slope km² /month |
|---|---|---|---|
| **Winter** | Increasing | 0.487 | 0.1142 |
| **Summer** | Decreasing | 0.976 | -0.0095 |





Table 6 Summary of SCA trend analysis from 2000 to 2020.

### 3.3.     Zone-wise Snow-Covered Area Analysis

**Elevation Zone (A).** Between 1774 and 3000 meters above sea level, covering a total area of 342.24 km² (3.91
percent). During the winter season, around 32% of this zone is covered in snow. Figures 11 represent the snow cover
dynamics of elevation zone (A) during summer and winter, respectively. The analysis reveals no significant trend in
snow cover during the summer season, while the winter session shows an increasing trend, consistent with previous
studies (Bilal et al., 2019; Atif et al., 2015)

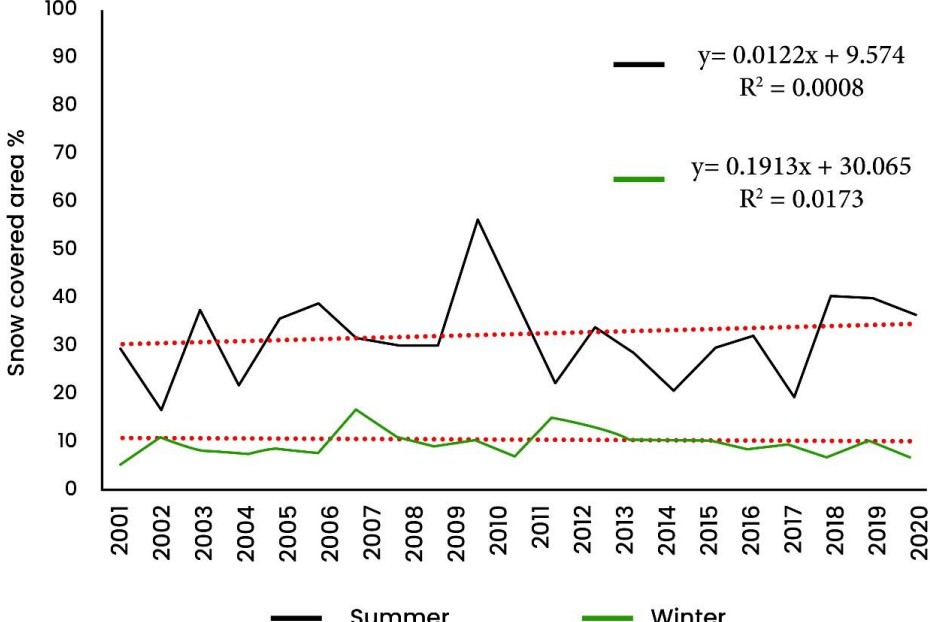

Figure 11 Snow cover variability in zone (A) during the winter season and summer session.

**Elevation Zone (B).** The zone (B) covers a total area of 2937.87 km² (33.64 percent). WAPDA has established only
one weather station in this zone, the Ziarat meteorological Station (3669 m). This zone has more persistent snow
cover as compared to zone (A) due to the higher elevation. According to (Bilal et al., 2019), a clear shift is
observable in snow accumulation and snowmelt periods in Zone (B) as compared to zone (A)   The SCA data
analysis of this zone also shows similar results as zone (A). The SCA during the summer season from 2000 to 2020
shows a linear trend while the SCA in the winter season shows an increasing trend. Tahir et al. (2011) the increasing





trends of precipitation continue to feed the high altitudes, particularly zone C, and result in the form of expanding
snow cover in the area. Figure 12 represents the trend of SCA in zone (B) during the summer season and winter
season, respectively.

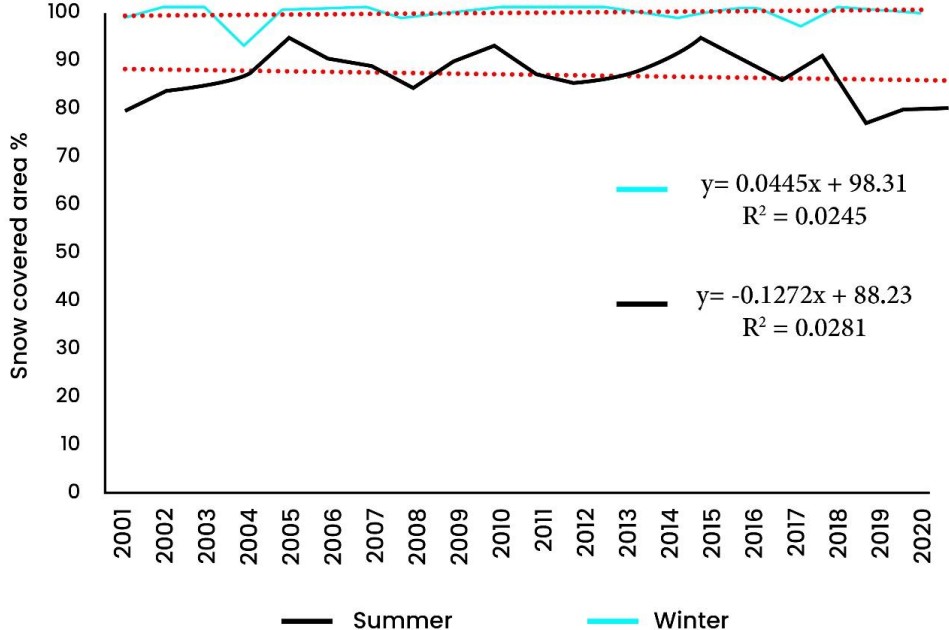

Figure 12 Snow cover variability in zone (B) during the summer season and winter session.
**Elevation Zone (C).** is the most central zone, with only one meteorological station (Khunjerab station). The annual
temperature ranges from -3°C in March to 10°C in August. SCA is more persistent in this zone than in zones (A) and
(B) due to the higher elevation and lower mean temperature. In this zone the SCA trend analysis revealed a
decreasing trend during the summer season, while the SCA trend during the winter season is non-significant and
linear to some extent, as represented in Figure 13. According to (Tahir et al., 2011) The tendency for snow
accumulation shows a marked increase during peak snow periods from November to February across all altitudinal
zones, with the most notable increase observed in the elevation above 4500 meters above sea level in zone C. During
the minimum snow cover periods from June to September, a slight upward trend is also apparent in zone C. A
similar non-significant increasing trend was also observed in the zone (C) (Tahir et al., 2015).



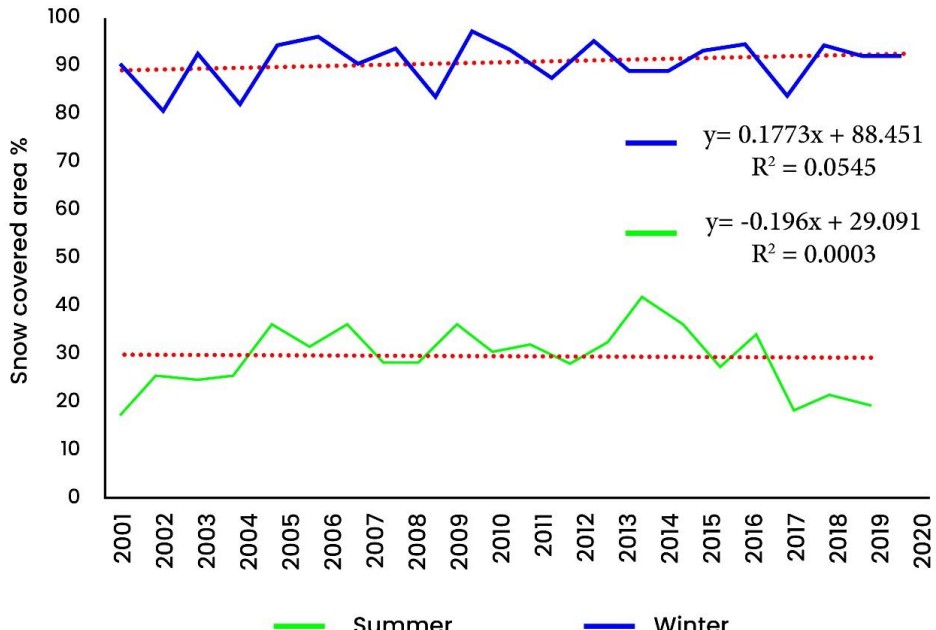

Figure 13 Snow cover variability in zone (C) during the summer season and winter session.

**Elevation Zone (D).** With a mean elevation of 6887 meters above sea level, Zone (D) is the highest elevation zone.
Throughout the year, more than 90% of the land is covered with snow (Bilal et al., 2019). Over the previous 20
years, analysis of monthly data revealed no trend during the winter season because the entire area of this region is
snow-covered during the winter season, while during the summer season, the snow-covered area persists because of
the higher elevation and low levels of temperature throughout the year as represented in figure 14. According to
(Bilal et al., 2019) SCA data further suggest significant increasing tendency during snow accumulation ($\tau$=+0.062,
p<0.05 S=+0.004% yr-1) and snowmelt ($\tau$=+0.181, p<0.05, S=+0.08% yr-1). Apart from the years 2016 and 2017,
SCA over this region, for the summer month shows homogeneity and a non-significant decreasing trend, as
represented in Figure 14.



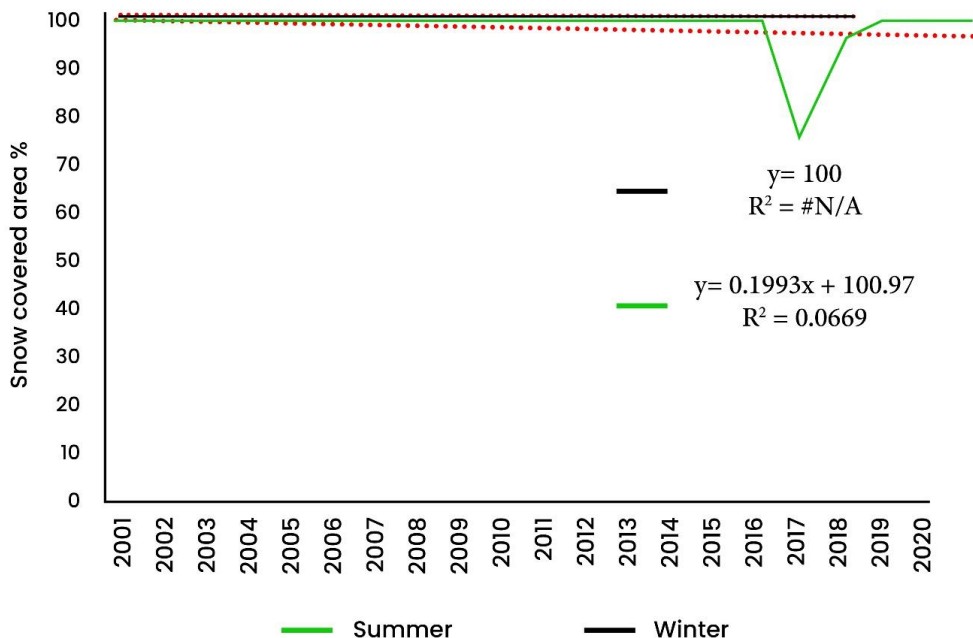

Figure 14 Snow cover variability in zone (D) during summer session and winter session

### 3.4.    Seasonal Temperature and Precipitation Correlations with SCA

Figures 15 and 16 represent the correlations of SCA variability along with the variations in the meteorological
parameters for both the winter and summer seasons respectively. The years depicting high SCA values are in good
agreement with low- temperature observations and high precipitation. During the winter season, a sudden increase in
precipitation was experienced after 2008-2009 which started to diminish from 2011. However, during the winters of
2002-2003, a peak in temperature was detected as compared to the other years, with a sudden dip in snow cover
during the year 2003 despite significant precipitation. The sudden fall in SCA during 2003 despite high precipitation
indicates that temperature has a stronger correlation with snow cover compared to precipitation. Similarly, another
slightly increase in temperature was observed during 2016, and the 2017-time period, for which the year 2017 also
shows a sudden dip in SCA. The overall analysis shows, that the snow cover during the winter season is not much
variable and has a non-significant increasing trend at a minute level (Figure 15).



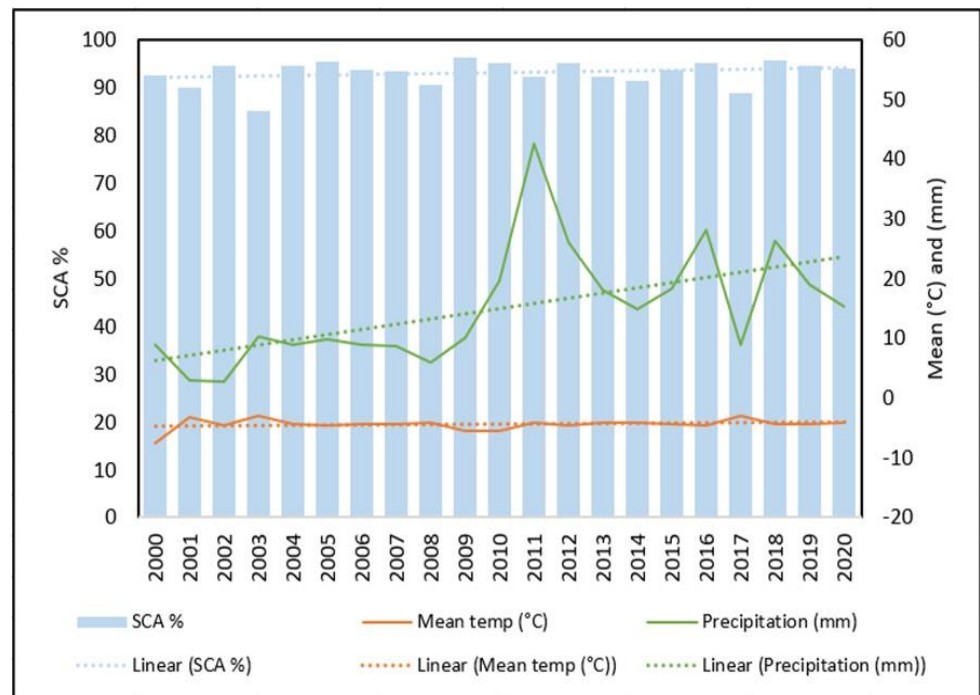

Figure 15 Snow cover variability in correlation with meteorological parameters during the winter season.

While, during the summer season, prior years 2014, 2004, and 2009 showed relatively more snow cover 6446.59
km², 6255.18 km², and 6138.27 km² respectively in an area on average which also correlates with the variations in
temperature and precipitation along the Hunza region. From Figure 16 it could be concluded that the year 2014 has
the maximum level of SCA because during the year temperature has a minute lag but the precipitation value has
risen, thus both these factors positively affect the snow cover resulting in the maximum level of SCA during this
year. The years 2018 and 2019 show less SCA as compared to other years because the temperature during these
years has slightly risen and the precipitation has also decreased. So, both these factors adversely affect the SCA
resulting in the low extent of SCA (Figure 16).



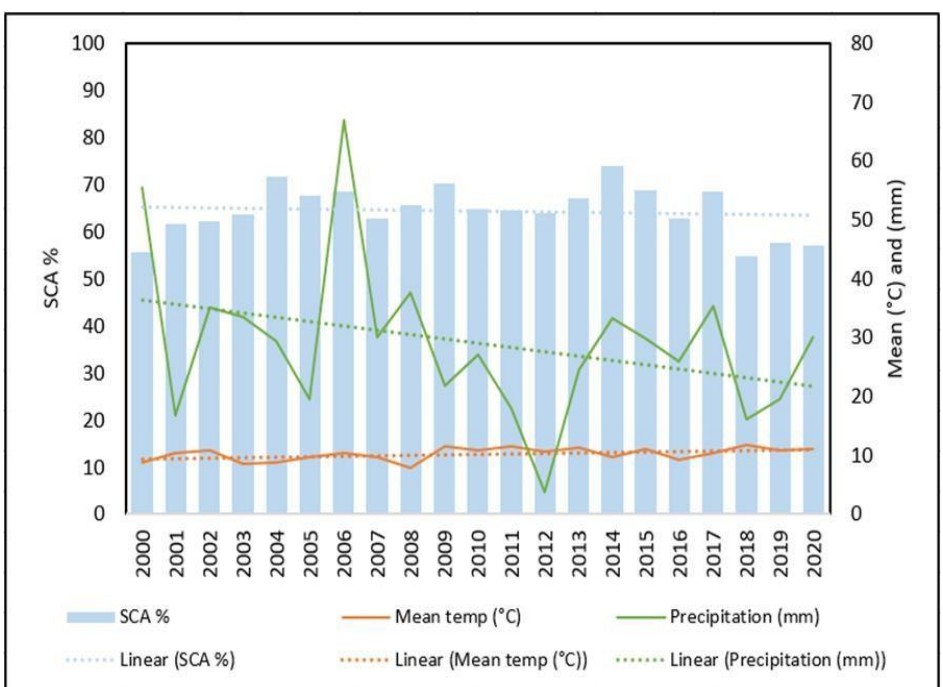

Figure 16 Snow cover variability in correlation with meteorological parameters during the summer season.

The strength of the SCA's correlation with temperature and precipitation was evaluated using Spearman's
Correlation Method. The analysis shows a significant, strong negative correlation between SCA and seasonal mean
temperature, with a correlation coefficient (r) of -0.804 for the winter season and -0.511 for the summer season over
the past 20 years (Table 7). This implies that SCA and temperature are inversely proportionated. Figures 15 and 16
illustrate that the enhanced SCA coincides with decreased temperature and increased precipitation over the Hunza
region.

|  | Winter | Summer |
|---|---|---|
| **Temperature** | -0.804 | -0.511 |
|  | P<0.0001 | P=0.620 |
| **Precipitation** | 0.371 | 0.182 |
|  | P=0.099 | P=0.428 |

Table 7 Spearman's Correlation coefficients between SCA (maximum extent) and climate variables over Hunza

during 2000–2020.



## 4. Summary

The Hunza sub-basin in the Upper Indus Basin (UIB) shows a substantial rise in snow-covered Area (SCA) above
4500m asl, standing in contrast to global snow cover patterns. Using MODIS satellite data (MOD10A2) and
meteorological data from 2000 to 2020, this study analyzed SCA trends and their relationship with temperature and
precipitation, supported by Mann-Kendall and Spearman correlation tests. The region experiences two distinct
seasonal snow periods: the summer season from April to August, during which the snow covers about 64% of the
area; and the winter season from September to March, during which snow covers about 93% of the total area. This
30-35% variation in snow-covered areas (SCA) contributes considerably to the continuous Indus River flow.
Generally, at the Sub-Basin Scale, the SCA of the Hunza region shows a non-significant advancing trend with a rate
of 0.114 km²/month during the winter season, while the SCA during the summer season shows a slightly decreasing
trend. This characteristic of SCA variability during both seasons can be attributed to the variations in the climatic
variables. During the winter season, the temperature trend of the Hunza region shows a minute level of increase with
a rate of 0.176 °C/month, but the amount of precipitation shows a significant increasing trend from 2000 to 2020,
which has ultimately resulted in the advancing trend of SCA during winter months. Apart from the significantly
increasing trend of precipitation, SCA is not progressing at a much greater extent during the winter season because
the temperature has a wide effect on SCA as compared to precipitation, canceling out the large increase in
precipitation. Similarly, during the summer season, the temperature trend shows a minute level of increase with a
rate of 0.31 °C/month, but at the same time, precipitation also has a decreasing trend that results in the decline of
SCA during this season. The precipitation levels during the winter increased due to Westerly circulation to the
Hunza region. The area receives precipitation in the form of snow due to its greater elevation, especially during the
winter season. The high elevation of the Hunza region, an increase in global precipitation, orographic precipitation,
the first encounter of winter westerly disturbances, increasing temperature, a shorter snow-melt season, and an
increase in relative humidity are all factors that contribute to the advancing snow cover and glaciers of the region.

### Data Availability

The MODIS 8-day snow data used in this study is freely accessible at: https://modis-snow-
ice.gsfc.nasa.gov/?c=MOD10A2. The Landsat images used in this study are freely available at:
https://earthexplorer.usgs.gov/. Meteorological data was obtained from the WAPDA meteorological station. This
data can be accessed by contacting the authors upon request

### Author Contributions

All authors contributed equally from the introduction to the conclusion under the supervision of the principal
investigator (PI). The study was completed through collaboration among all authors. Dr. Garee Khan supervised the
study and helped with graphs, maps, and manuscript writing, Mr. Muqeet Ahmed developed the main research
concept, conducted the study, and worked on graphs, maps, and manuscript writing, Mr. Shehzad Ali contributed to



the manuscript writing and conducting the study, Ms. Parisa Karim and Mr. Muzammil Hassan assisted in field data
collection and research and Mr, Kelden Jurmey, Ms, Lhachi Dema, and Ms. Shrija Gurung provided relevant data
analysis, and reviewed, and edited the manuscript before submission.

**Competing Interests**

The authors declare that they have no conflict of interest.

**Disclaimer**

The authors declare that the views, interpretations, and conclusions presented in this study are solely their own and
do not necessarily reflect the views or policies of their affiliated institutions. The data, methodologies, and analyses
used in this research are based on publicly available sources and standard scientific techniques; however, the authors
assume no liability for any errors or omissions. The findings are intended for academic and research purposes only
and should not be used for decision-making without further validation.

**Acknowledgements**

We acknowledge the National Aeronautics and Space Administration (NASA) for making MODIS snow cover
products freely available, and the United States Geological Survey (USGS) for providing Landsat imagery. Special
appreciation is extended to the Water and Power Development Authority (WAPDA) for providing meteorological
data. The guidance and support from Dr. Garee Khan and the contributions of all co-authors in data analysis,
fieldwork, and manuscript preparation are highly appreciated.

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
