# Peer review of "Analysis of Snow Cover Changes Using MODIS Snow Products and Meteorological Data in the Hunza Region, Karakoram, Pakistan"

_EGUsphere, 2024_

## Author Comment (AC1)

**Response to Main Comments**

We appreciate your review and feedback on our manuscript. We have made all the minor changes suggested. Please find below the remaining points addressed. Our answers are in blue.

1.  The first critical point regards the English, which is very poor. Several sentences have been written without a verb (e.g. pag. 11, Lines 172-173; pag. 16, Line 228). In addition, many sentences and concepts are redundant. There are also several typos, sometimes the acronyms are used before they have been explicitly stated, and the references are cited in the main text with a non-uniform style.

**Response**: Thank you for the valuable feedback. I have thoroughly revised the entire manuscript to improve the English language, ensuring that all sentences are grammatically correct and complete. Redundant phrases have been removed, acronyms are now explicitly defined upon first use, and the citation style throughout the text has been made consistent, following the required guidelines.

2.  From a methodological perspective, there are several "dark points" that should be clarified:

**Line 99**: A very poor method to reconstruct missing data in a time series.

**Response**: Thank you for highlighting this important point. We acknowledge that using Terra and Aqua MODIS data solely for the purpose of removing missing snow cover pixels on specific days is a simplified approach and may not fully capture the temporal variability of snow cover. Our primary goal in this step was to ensure data completeness for accurate trend analysis, given that missing data points can introduce biases or gaps in the time series.

While this method may be considered limited, it was chosen due to its straightforward implementation and the high temporal resolution of MODIS 8-day composites, which help minimize data gaps. We recognize that more advanced gap-filling techniques, such as interpolation methods, temporal smoothing, or modeling approaches, could provide a more robust reconstruction of missing data and better preserve the underlying temporal dynamics.

In future work, we plan to incorporate such sophisticated data reconstruction methods to improve the continuity and reliability of the snow cover time series, thereby strengthening the overall analysis. We appreciate this constructive feedback and will consider it to enhance the methodological rigor of our work.

**For Temperature and Precipitation Missing Data**

Similarly, to address the missing data in the temperature and precipitation time series, we used the temperature and precipitation gradient techniques. These methods, commonly used in meteorology and climatology, estimate missing values by analyzing the typical variations of temperature and precipitation across neighboring locations or time periods.

Specifically, we applied these gradient-based techniques to fill in spatial or temporal gaps in the data. Additionally, we used interpolation from the preceding and following years to maintain consistency and continuity in the overall analysis.

**Line 94:** It is necessary to provide additional details about the method used to classify the Landsat images.

**Response:** We collected Landsat images to get a clear snapshot of the area of interest (AOI), as they provide a detailed background and accurate elevation data. Using these images, we divided the AOI into different elevation zones. Landsat Images are used to make the study area boundary.

**Line 136**: Why 8-day interval images? Can you justify this choice?

**Response**: The MODIS/Terra 8-day snow product (MOD10A2) is widely used in modern research for snow cover analysis. These are the following reasons why we selected MODIS:

The MODIS/Terra 8-day snow product (MOD10A2) was selected because it contains the data of High Mountain Asia (HMA) covering latitude 24.32 − 49.19 N and Longitude 58.22 − 122.48 E

The MODIS/Terra 8-day snow product (MOD10A2) was selected for its ability to mitigate cloud obscuration by retaining the maximum snow extent observed over the period, which is critical in the Hunza region where persistent cloud cover often hinders daily observations (Riggs et al., 2006).

This product optimally balances temporal resolution (8 days) with spatial detail (500 m), ensuring consistent long-term trend analysis (2000–2020) while minimizing gaps from daily noise (Ahmad et al., 2018).

Its reliability for snow cover mapping in the Upper Indus Basin has been validated by prior studies such as (Tahir et al., 2015; Hasson et al., 2014), reporting errors of <10% in high-elevation zones. Additionally, the 8-day composites reduce data volume without compromising seasonal snow dynamics, aligning with the study's focus on monthly trends and operational feasibility.

**References**

Ahmad, I., Ahmad, Z., Munir, S., Shah, S. R. A., and Shabbir, Y.: Geo-spatial dynamics of snowcover and hydro-meteorological parameters of Astore basin, UIB, HKH Region, Pakistan, Arabian Journal of Geosciences, 11, 1-15, https://doi.org/10.1007/s12517-018-3734-9, 2018.

Hasson, S., Lucarini, V., Khan, M. R., Petitta, M., Bolch, T., and Gioli, G.: Early 21st century snow cover state over the western river basins of the Indus River system, Hydrol. Earth Syst. Sci., 18, 4077-4100, 10.5194/hess-18-4077-2014, 2014.

Riggs, G. A., Hall, D., and Salomonson, V.: MODIS snow products user guide, NASA Goddard Space Flight Center Rep, 80, 1-45, 2006.

Tahir, A. A., Chevallier, P., Arnaud, Y., Ashraf, M., and Bhatti, M. T.: Snow cover trend and hydrological characteristics of the Astore River basin (Western Himalayas) and its comparison to the Hunza basin (Karakoram region), Science of the total environment, 505, 748-761, https://doi.org/10.1016/j.scitotenv.2014.10.065, 2015.

3. Figure 4 does not provide clear information about the altitudinal zones extracted from GDEM. It is not clear which is the altitudinal range of the elevation zones B and C.

**Response:** The altitudinal ranges and details of zones B and C are explicitly provided in Table 2 (Page 5) and the hypsometric curve (Figure 2, Page 5). Here's the clarification:

Elevation Zones in the Hunza Region

**Zone B**:

Elevation range: 3,000–4,500 meters above sea level (m asl).

Mean elevation: 3,750 m asl.

Land area: 2,937.87 km² (33.64% of the study region).

**Zone C** :

Elevation range: 4,500–6,000 m asl.

Mean elevation: 5,250 m asl.

Land area: 5,328.04 km² (61.01% of the study region).

Zone C dominates the basin (61% of the area) and is critical for snow persistence due to temperatures below freezing year-round (Page 15, Lines 205–207). Mostly, the population lives in Zone B and Zone C because it's a mountainous region, as mentioned above.

4. The scientific quality of the analysis is poor and is limited to a simple and basic discussion of the trend of SCA and meteorological variables, without an in-depth investigation into other potential drivers of SCA variability (e.g., teleconnection patterns). In this sense, additional efforts should be carried out.

**Response:** Thank you for your valuable feedback. The primary objective of our study was to provide an initial assessment of snow cover area (SCA) trends using MODIS 8-day Terra and Aqua data, which are widely recognized for their high temporal resolution and broad spatial coverage. Our focus was to establish baseline patterns and temporal trends in SCA over the study region, which we believe is a necessary step before conducting more complex analyses.

We acknowledge that a comprehensive understanding of SCA variability involves examining additional drivers, such as teleconnection patterns and larger-scale climate oscillations. However, the scope of this study was intentionally limited to the analysis of direct snow cover trends and their correlation with meteorological variables due to data availability, computational resources, and to maintain the clarity of the initial findings.

Future research, as suggested, will aim to incorporate these additional factors, including teleconnection indices, to deepen the analysis of drivers influencing SCA variability. We appreciate this constructive suggestion and agree that such efforts would significantly enhance the scientific robustness of the study.

5. Finally, I suggest strongly improve the quality and style of the figures

**Response:** We have updated and changed the figures and added all figures with high quality and style to the manuscript as per your instructions

[Figure]

Figure 2 Hypsometric curve of the Hunza region estimated through digital elevation model (DEM).

[Figure]

Figure 4 shows the altitudinal zones of the Hunza extracted from the Global Digital Elevation Model (GDEM) along with glacier coverage and gauging stations.

[Figure]

Figure 5 Temperature trend in the Hunza region during both summer and winter seasons from 2000 to 2020.

[Figure]

Figure 6 Precipitation trend in the Hunza region during both summer and winter seasons from 2000 to 2020.

[Figure]

Figure 7 Spatial variability of snow cover during the summer season from 2000 to 2020.

[Figure]

Figure 9 Spatial variability of snow cover during the winter season from 2000 to 2020.

[Figure]

Figure 16 Snow cover variability in correlation with meteorological parameters during the summer season.

**Response to Minor Comments**

**Line 9:** The acronym SCA has not been explicitly stated.

**Response**: Snow-Covered Areas

**Line 36:** The acronym HKH has not been explicitly stated.

 **Response:** Hindu-Khush Himalayas

**Line 30:** "according to the Intergovernmental Panel on Climate Change (IPCC) report titled "State of the Global Climate 2020 WMO-No. 1264", between 1880-2020 where the average land and ocean temperature has increased globally by 1.2°C." This sentence should be carefully revised.

**Response**: According to the Intergovernmental Panel on Climate Change (IPCC) report titled "State of the Global Climate 2020 WMO-No 1264", the average global land and ocean temperature increased by 1.2°C between 1800 and 2020.

**Line 53:** "Immerzeel et al. (2009) the same pattern was observed between 2000 and 2008." Please carefully check the English grammar.

**Response:** A similar pattern was observed between 2000 and 2008 by Immerzeel et al. (2009).

**Line 67:** The acronym UIB has been just explicitly stated at this point of the manuscript.

**Response**: Upper Indus Basin

**Line 99:** What is the percentage of missing data?

**Response**: Almost 25%

**Line 107:**" Winter (March) and Summer (August)". Season should be from a determined month to another month.

**Response**: Winter: September- March and Summer: April- August

**Line 110:** The acronym MK is not explicitly stated;

**Response**: Mann-Kendall

**Line 121:** "is defined using the using the". Please revise this sentence.

**Response**: I have deleted the double using the

**Line 125**:" data at a resolution of 30-meter resolutions". Please carefully revise this sentence.

**Response**: Data at a 30-meter resolutions.

**Line 125**: To address these voids in the STRM DEM were filled using the fill function in ArcMap". This sentence is not clear.

**Response:** To address these voids, the STRM DEM was processed using the 'Fill' tool in ArcMap 10.5, which fills missing elevation values in the dataset.

**Line 126:** "data was". Please check the English grammar.

**Response:** Data Were

**Line 129**: Figure 2.2?

**Response:** Figure 2

**Line 133**: The equations should be carefully revised. A detailed description of the parameters embedded in them is missing.

**Response:** Where:

**S:** is the Mann-Kendall test statistic that measures the trend direction.

**n:** is the number of observations.

$f_i$: is the number of data points in the $t$-th tied group.

**Var:** is the variance of the statistic $S$, adjusted for ties.

**se**: Standard error of **S**, usually calculated from the variance of S.

**z**: Standardized test statistic (Z-score), which helps determine the significance of the trend:

**Tables 2 e 3** are not mentioned in the main text.

Response: Thank you for your valuable feedback. We apologize for the oversight regarding the mention of Table 2 and 3 in the main manuscript. we have now updated the text to include references to these tables at the relevant sections in the manuscript.

1. For Table 2 (Hypsometric Zones of the Hunza Region):

Add text at line 89 (in the "Methodology" section, where the discussion about the challenges due to the high altitude and low altitude data gap begins).

2. For Table 3 (Summary of the Datasets Used in the Meteorology data):

Add text at line 145 (in the "Materials and Methods" section, after discussing the datasets).

**Line 153:** What represent the red and blue lines in this figure? An average between the two stations? Please clarify.

In addition, please uniform the style of this figure to that of Figure 6.

**Response:** From the Figure, the red and blue lines represent linear trend lines drawn over the average seasonal mean temperature for winter and summer, respectively, based on the combined data from the Khunjerab and Ziarat meteorological stations.

**Lines 172-177:** this concept is not clear and the sentences seem to be not adequately connected among each other.

**Response**: In the Upper Indus Basin (UIB), seasonal snow-covered area (SCA) is strongly influenced by winter precipitation Bilal et al. (2019) and Atif et al. (2018). The Increasing trend in winter precipitation observed in this study aligns with previous findings by Tahir et al. (2011), who reported an expansion in snow cover in the Hunza basin as a result of higher winter precipitation. Additionally, (Fowler and Archer, 2006) noted that between 1961 and 2000, the Karakoram Mountains experienced decreasing summer temperatures and a significant increase in overall precipitation. These climatic changes have likely contributed to the persistence or growth of snow cover in the region.

**Line 186**: "The trend analysis of SCA during the summer season is shown in Figure 8 SCA analysis during the summer season represents a non-significant declining trend for the period of 2000 to 2020". Please carefully revise this sentence.

**Response**: Figure 8 represents the trend analysis of SCA during the summer season, which reveals a non-significant declining trend for the period 2000-2020.

**Line 198**: "The trend analysis of SCA during the winter season is shown in Figure 10. SCA analysis during the winter season represents a minute, non-significant increasing trend for the period of 2000 to 2020". Please carefully revise this sentence.

**Response**: Figure 10 presents the trend analysis of SCA during the winter season, which shows a minute, non-significant increasing trend from 2000 to 2020

**Line 201**: The SCD trend in winter is not significant, as the authors state in the Abstract. Please carefully check.

**Response**: Non-significant

**Lines 207-210**: "32.5% of the Hunza Basin region is located above 5,000 meters above sea level, and the seasonal variation in snow cover ranges from 38 to 43%. At this altitude, temperatures remain below freezing throughout the year and precipitation tends to increase. This may be the reason for the increase in snow cover in the Hunza basin and the advance of some glaciers." This concept has been introduced in the previous Line.

**Response**: We have deleted the repeated sentence that was added mistakenly

**Line 210:"** (Hewitt, 2005; Archer and Fowler, 2004) also concluded". Please revise this sentence.

**Response**: Hewitt (2005) and Archer and Fowler (2004)

**Line 218**: "Figure 11 represents". Please check the English grammar.

**Response**: Figure 11 shows the snow cover dynamics in elevation zone (A) during the winter and summer seasons.

**Lines 233-241**: Please carefully revise such sentences.

**Response**: is the central and highest-coverage zone, monitored by only one meteorological station (Khunjerab station). The temperature here ranges from approximately -3°C in March to 10°C in August. Due to its higher elevation and cooler climate, this zone shows more persistent snow cover compared to Zones (A) and (B). The trend

analysis indicates a decreasing SCA trend in summer, while the winter SCA trend is generally stable and non-significant, as shown in Figure 13. According to Tahir et al. (2011), snow accumulation increases notably during the peak winter months (November to February), especially in areas above 4500 meters, such as Zone (C). Additionally, a slight increase in SCA during the low-snow period (June to September) was also observed. Similar non-significant but upward winter trends in this zone were reported by (Tahir et al., 2015)